# Exploring the Association Between Problematic Internet Use, Internet Gaming Disorder in Adolescents with ADHD: A Scoping Review

**DOI:** 10.3390/ijerph22040496

**Published:** 2025-03-26

**Authors:** Roberto Ghiaccio, Anna Passaro, Fabrizio Stasolla, Elvira Martini, Angelo Maria De Fortuna, Raffaele De Luca Picione

**Affiliations:** 1Faculty of Law, Giustino Fortunato University of Benevento, 82100 Benevento, Italy; r.ghiaccio@unifortunato.eu (R.G.); f.stasolla@unifortunato.eu (F.S.); e.martini@unifortunato.eu (E.M.); r.delucapicione@unifortunato.eu (R.D.L.P.); 2Department of Communication Sciences, Humanities and International Studies (DISCUI), University of Urbino, 61029 Urbino, Italy

**Keywords:** ADHD, adolescence, internet gaming disorder, problematic use disorder, comorbidity

## Abstract

Background: Attention-Deficit/Hyperactivity Disorder (ADHD) is a neurodevelopmental condition characterized by inattention, hyperactivity, and impulsivity. Adolescents with ADHD have an elevated risk of developing Internet Gaming Disorder (IGD), a condition involving excessive gaming that disrupts daily life. IGD is linked to traits such as low frustration tolerance and sensation-seeking, with comorbid conditions like anxiety and depression further increasing vulnerability. Gaming frequently serves as a coping strategy due to emotional regulation difficulties. The dynamics within family units and peer relationships play a pivotal role, with dysfunctional environments heightening the risks and positive interactions serving as protective factors. Methods: This scoping review analyzed empirical studies published in the last decade exploring the association between ADHD, Problematic Internet Use (PIU), or IGD, focusing on neurobiological, psychological, and environmental factors. Results: The findings highlight that impulsivity and emotional dysregulation in ADHD contribute to IGD. Gaming is frequently used as a maladaptive coping strategy, with social and family influences modulating risk. Diagnostic complexities arise in distinguishing ADHD-related behaviors from IGD symptoms. Conclusions: Addressing these comorbid conditions requires interdisciplinary collaboration and evidence-based interventions. Future research should focus on understanding ADHD, PIU, or IGD interactions and developing targeted interventions. Longitudinal studies are necessary to establish causal links and assess effective treatment strategies.

## 1. Introduction

Adolescence is a crucial period in human development in which mental health problems often emerge due the rapid physical growth, hormonal changes, and emotional development [1]. Adolescents are integrating digital technologies into their daily lives, as evidenced by the popularity of video games and Internet use. A prevailing issue among adolescents is that of compulsive Internet use, which has been demonstrated to be associated with a number of emotional processing difficulties, including alexithymia and dissociative symptoms [2].

Qualitative research has yielded nuanced insights into adolescents’ perceptions of Problematic Internet Use, elucidating their interpretations, coping strategies, and justifications for excessive online behaviors [3]. The significance of adolescents’ subjective experiences is further emphasized by findings indicating motivational aspects in retrospective evaluations of school experiences [4].

Given the socialization, entertainment, and educational opportunities these platforms provide, they have become an essential component of contemporary adolescence [5]. Problematic Internet Use (PIU) and Internet Gaming Disorder (IGD) are increasingly prevalent among adolescents. PIU is characterized by a lack of control over Internet use, leading to negative consequences in daily life. Persons with PIU or problematic online gaming often have difficulty setting limits to the amount of time they spend online, which can seriously hamper their social and emotional wellbeing as well, and can be deleterious for their ability to fulfill obligations to others, their achievements, or their jobs [6,7]. This phenomenon is referred to by various terms, such as “Internet addiction”, “problematic video gaming”, “video game addiction”, and “problematic Internet gaming” [8]. These labels draw attention to specific manifestations of PIU, particularly in online gaming communities, where social connectivity and immersive environments may exacerbate overuse.

Despite the presence of commonalities in the behavioral manifestations of PIU and IGD, these conditions are conceptually distinct. PIU is defined as excessive Internet use that leads to a variety of psychosocial problems, including impairments in academic, occupational, and social relationships. It encompasses both generalized excessive online behaviors and specific problematic uses, such as social media addiction, online pornography consumption, compulsive browsing, excessive gaming, online gambling, and smartphone dependency [9,10]. IGD is characterized by a preoccupation with gaming, withdrawal symptoms when gaming is not possible, tolerance, and the continuation of gaming activities despite negative consequences [11,12]. In contrast, PIU does not necessarily entail gaming; it may also encompass a variety of online platforms or activities, each of which has the capacity to interfere with daily functioning. This distinction underscores the necessity to distinguish IGD as a distinct subtype of PIU, given that gaming activities frequently provide structured, immediate feedback and social reinforcement. Accordingly, the present review includes both PIU and IGD to capture the full range of problematic online behaviors observed in adolescents with ADHD.

The co-occurrence of Attention Deficit Hyperactivity Disorder (ADHD) and IGD has been investigated, showing a significant overlap between the two disorders [13,14,15]. Persons with ADHD are at increased risk of developing addictive behaviors, including Internet addiction and IGD, due to difficulties with time management, task prioritization, and hyperfocus [16,17,18]. In addition to cognitive factors, the psychosocial challenges experienced by adolescents with ADHD further increase this risk. ADHD is a persistent neurodevelopmental disorder characterized by inattention and hyperactivity/impulsivity, which can significantly impact daily life [11]. Among the hallmarks in ADHD is impulsivity, which is characterized by a tendency to act without planning or considering the consequences. When combined with the rewarding nature of video games, impulsivity produces a powerful feedback loop that reinforces challenging behaviors. The reward system and social support provided by video games can intensify the severity of PIU and IGD among adolescents with ADHD [19,20]. Adolescents with ADHD may turn to video games as a coping strategy for their personal and academic difficulties, frequently using them to satisfy their need for stimulation and to reduce feelings of loneliness [21,22]. This reliance on gaming can lead to problematic behaviors, specifically when combined with attention and impulse control deficits associated with ADHD. According to the literature [23,24], the structured format of games and their immediate feedback create a sense of achievement, making them particularly attractive to persons with ADHD. The neurobiological mechanisms of ADHD, such as the imbalance between brain regions involved in reward processing and cognitive control, may explain why adolescents with ADHD are more prone to problematic social media use. Social media provides an experience of immediate gratification, which may be particularly appealing to persons who struggle with impulse control [25]. Furthermore, it has been demonstrated that adolescents who grow up in dysfunctional families—characterized by poor communication, high levels of conflict, and inconsistent parenting—have been shown to be more prone to Problematic Internet Use and IGD [26,27]. Adolescents with ADHD are especially vulnerable in these environments, as they frequently have difficulty regulating their emotions and are more likely to turn to unhealthy coping mechanisms such as excessive gaming or Internet use [28]. In addition, adolescents with ADHD have more severe social phobia and engage in certain online activities, such as playing online games, using e-mail, or participating in social networks, and may face an increased risk of developing PIU [29]. The aim of this scoping review was to systematically explore and map the existing literature on the relationship between PIU, or IGD, and adolescents diagnosed with ADHD. Given the increasing digital engagement of adolescents, this review aimed to identify key patterns, gaps, and findings in the research to better understand how these conditions intersect. The review focuses on adolescents, highlighting a crucial developmental phase when online gaming practices are most pronounced and providing specific insights for this age group. Ultimately, by emphasizing the development of prevention strategies and targeted treatments for PIU and IGD in adolescents with ADHD, this review aims to offer a comprehensive synthesis of current evidence to guide research and clinical practice.

## 2. Methods

The scoping review was carried out according to the guidelines Preferred Reporting Items for Systematic Reviews and Meta-Analyses (PRISMA; see Figure 1) [30]. The research was conducted using the Scopus database. The search strategy involved combining specific keywords to identify relevant studies. The terms included “ADHD” and “adolescence”, combined with either “problematic use disorder” or “Internet Gaming Disorder”. Initially, the titles and abstracts of the studies were screened to determine their correspondence with the eligibility criteria. Subsequently, articles deemed potentially relevant underwent to a full-text review.

Eleven studies that met predetermined eligibility criteria were reviewed and forty pertinent records were selected. The eligibility criteria are detailed below.

### 2.1. Inclusion Criteria

Studies meeting the following criteria were included:a.studies published within the interval range 2014 to 2024;b.population of interest is adolescents;c.the study design includes empirical research, using quantitative, qualitative, or mixed approaches;d.the aim of the studies must explore the association between Problematic Internet Use and/or Internet Gaming Disorder and attention deficit/hyperactivity disorder; only studies written in English were considered.

### 2.2. Exclusion Criteria

Studies were excluded if they had the following characteristics:a.adults or children under the age of 10;b.review articles, commentaries, books, editorials, or letters;c.studies published in languages other than English; records not relevant to the research aim.

## 3. Results

Özkan et al. [16] found that problematic social media use, symptoms of ADHD, and adolescent experiences of being a victim of cyberbullying were positively correlated.

The study utilized multiple linear regression analyses to demonstrate that social media addiction and cognitive monitoring significantly exacerbate inattention and cognitive impairments in adolescents with ADHD.

Similarly, Shuai et al. [31] identified that in comparison to their ADHD peers who do not use problematic digital media (PDMU), adolescents with ADHD who use PDMU present more severe attention deficits, emotional and behavioral difficulties, and worse academic results.

Jeong et al. [32] highlighted several risk factors for developing IGD, including extended gaming during weekdays, a preference for multiplayer games, symptoms of depression, anxiety, and ADHD, weaker parental attachment, reduced openness in parent-child communication, and limited social support.

A further study [33] validated a psychometric tool the IGD-SF-T-L scale for diagnosing IGD in Taiwanese adolescents with ADHD. The researchers’ findings supported the scale’s reliability, recommending a diagnostic cutoff score of 10 for IGD.

The study’s findings supported the scale’s reliability, recommending a diagnostic cutoff score of 10 for IGD.

Hsieh et al. [34] analyzed how mental health, parenting styles, and adolescent behaviors influence parental self-efficacy in managing Internet use in adolescents with ADHD. They found that higher levels of oppositional defiant disorder (ODD) symptoms and Internet addiction in adolescents were linked to lower parental self-efficacy.

Hygen et al. [35] investigated bidirectional relationships between IGD and symptoms of other psychiatric disorders in adolescents. The results obtained demonstrated a significant association between an increase in IGD symptoms and a decrease in anxiety symptoms 2 years later.

Concurrently, Seyrek et al. [36] observed a significant association between Problematic Internet Use and the presence of depression, anxiety, and ADHD symptoms in adolescents. In their study, 16.2% of participants were identified as having probable IA, while 1.6% were diagnosed with IA.

Gostoli et al. [37] examined the relationship between ADHD symptom severity and Problematic Internet Use in Italian adolescents, finding that more severe ADHD symptoms correlated with greater Internet use problems. Notably, 22% of the sample exhibited subclinical ADHD, while an additional 22% were diagnosed with clinical ADHD.

Another study [17] investigated the influence of psychosocial factors, such as psychological wellbeing, on the relationship between ADHD and the use of technological devices in adolescents. The study concluded that high levels of specific psychological wellbeing (personal growth and purpose in life) may increase the impact of ADHD symptoms on time spent with smartphones.

Benedetto et al. [38] assessed the connection between smartphone distraction and ADHD-related symptoms in adolescents, reporting that greater smartphone distraction was associated with increased hyperactivity, inattention, and emotional difficulties. Finally, Lung et al. [39] investigated the role of dissociative absorption trait in the relationship between ADHD diagnosis and duration of Internet use. The results of this study indicated that the adolescents diagnosed with ADHD showed a higher dissociative absorption trait, which increased the number of hours spent engaged in online activities.

A salient finding across multiple studies is that adolescents diagnosed with ADHD exhibit a substantially elevated risk and severity of problematic digital behaviors in comparison to their peers without ADHD, encompassing both IGD and broader PIU phenomena such as problematic social media use and smartphone distraction [16,31,36,37,38].

Across a range of cultural contexts, including Italy [16,37,38] Turkey [36], South Korea [32] Taiwan [33], and China [31], ADHD has been shown to be a consistent predictor or correlational factor associated with an elevated risk of both IGD and broader PIU behaviors. A number of critical differences in study methodologies and diagnostic approaches have been identified. For instance, studies such as those by Chang and Tzang [33] and Jeong et al. [32] have explicitly focused on IGD, employing specific psychometric tools grounded in DSM-5 criteria (IGD-SF-T-L and IGUESS, respectively), enhancing diagnostic precision and allowing for clearer identification of pathological gaming behaviors. Conversely, studies such as those by Benedetto et al. [38], Shuai et al. [31], and Seyrek et al. [36] used broader measures (e.g., Young’s Internet Addiction Test) to capture a wider spectrum of problematic Internet activities, including social media addiction, smartphone distraction, and compulsive browsing behaviors. Furthermore, methodological heterogeneity is evident. Seyrek et al. [36], for instance, employed a cross-sectional design to explore correlations among IA, depression, anxiety, and ADHD symptoms in a sample of Turkish adolescents. This study demonstrated a significant association between IA and multiple psychopathological conditions, including depression, anxiety, and ADHD symptoms. However, it is notable that the study did not differentiate between specific subtypes of Internet activities. Conversely, longitudinal studies by Jeong et al. [32] and Lung et al. [39] have provided more robust data on causality and temporal stability, highlighting ADHD symptoms as strong predictors for persistent IGD or PIU over time. Specifically, Jeong et al. [32] found that prolonged daily gaming (>4 h) combined with ADHD symptoms significantly predicted both onset and persistence of high-risk IGD. Conversely, Lung et al. [39] identified dissociative absorption traits as mediating factors influencing PIU severity among adolescents previously diagnosed with ADHD. Despite the extensive nature of PIU, Seyrek et al. [36] observed associations specifically with behavioral components of ADHD, such as hyperactivity and attention deficits, which is consistent with the findings of other studies [16,38]. This finding suggests the presence of a shared vulnerability related to impaired impulse control and attentional dysregulation across various forms of PIU and IGD. However, the reviewed studies generally did not adequately differentiate between cognitive and motor impulsivity, which limits clarity regarding which specific subtypes of impulsivity might differentially influence IGD versus general PIU behaviors.

Shuai et al. [31] and Hsieh et al. [34] emphasized the detrimental impact of problematic digital media use on family cohesion, parental self-efficacy, and adolescents’ emotional regulation. In a similar vein, Seyrek et al. [36] indirectly corroborated these findings by highlighting elevated behavioral issues among adolescents with IA, which could potentially intensify parenting challenges and familial dynamics. Furthermore, recent literature [40,41] suggests that targeted parental mediation strategies significantly mitigate adolescents’ problematic Internet behaviors. However, none of the reviewed studies explicitly evaluated the effectiveness of such interventions.

Few studies [16,36,37] included a cross-sectional design, collecting data at a single time point to evaluate the associations between variables. Conversely, further studies [32,35], on the other hand, adopted a longitudinal approach, following participants over time to examine the progression of behaviors and the dynamics of relationships between variables over the years (see the Table 1).

## 4. Discussion

The reviewed studies highlighted the complexity of interactions between ADHD, problematic social media and Internet use, cyberbullying, and other psychosocial factors in adolescents. Notably, adolescents with ADHD exhibit higher levels of dysfunctional metacognitions, problematic social media use, cyberbullying, and cyber-victimization compared to their peers without ADHD [16]. The phenomenon of cyberbullying among adolescents is also indicative of broader sociocultural contexts, which serve to shape the manner in which young people interpret their experiences in the online sphere [42].

The applications of regression analysis revealed positive associations between ADHD symptoms, specific metacognitions (e.g., metacognitive worry and cognitive monitoring), and problematic social media use. Interestingly, cognitive monitoring was negatively associated with inattention symptoms and overall ADHD severity, suggesting that lower cognitive monitoring was linked to more pronounced symptoms [16]. Adolescents with ADHD commonly experience social difficulties, such as peer rejection and victimization, which can lead to seeking connections online [25]. Shuai et al. [31] investigated the impact of PDMU on adolescents with ADHD during the COVID-19 pandemic. The study demonstrated that ADHD adolescents with PDMU exhibited worse core symptoms, heightened emotional distress (e.g., anxiety and depression), and impaired executive function compared to those without PDMU. Furthermore, PDMU exacerbated family dynamics, increasing conflict and reducing cohesion. Additionally, it emerged that dissociative traits, such as absorption, mediated the relationship between ADHD and PIU [39,43]. Findings suggested that persons with ADHD may use the Internet to escape negative emotions, but doing so may reinforce maladaptive coping mechanisms. The outcomes underscored the need for early interventions targeting these traits to prevent the progression of PIU.

Adolescent Internet addiction has been shown to be significantly predicted by individual vulnerability factors, such as internalizing and externalizing problems, especially hyperactivity and inattention, as well as cognitive-emotional processes, such as emotion regulation, attention impulsivity, online vigilance, and multitasking [38].

The literature suggested that difficulties in emotional regulation, inhibition, and working memory may contribute to vulnerability to PIU or IGD [31,32]. Adolescents with ADHD frequently struggle with behavioral and emotional regulation issues, which can leave parents feeling frustrated and uncertain about how to manage their child’s Internet use effectively. This frustration may undermine their confidence and ability to set boundaries, creating a challenging dynamic. Accordingly, the adolescents’ Problematic Internet Use can further reduce parents’ sense of self-efficacy, exacerbating a vicious cycle that becomes increasingly difficult to break down [34]. The researchers recommended that interventions aiming to improve these cognitive functions may help reduce the severity of IGD symptoms [44].

Literature analysis noted the validation of the IGD-SF-T-L, which is specific to adolescents with ADHD. Their work highlighted the particular vulnerability of adolescents with ADHD to IGD due to emotional dysregulation, impulsivity, and addiction to gaming as a coping mechanism. This scale provides a valuable tool for clinicians to identify and address IGD early in ADHD populations [33].

Studies highlight the importance of early identification of ADHD symptoms, including subclinical ones, and related issues (e.g., problematic use of digital media) [16,17].

Additionally, studies showed the importance of providing training and support to parents and caregivers working with adolescents with ADHD. This support can improve the management of ADHD symptoms, the use of the Internet and digital media, and strengthen the parent-child relationship [17,34,45]. Moreover, recent studies have emphasized the role of parental self-efficacy, demonstrating how involvement in intervention programmers positively impacts quality of life in families managing complex developmental conditions [46]. These strategies have been shown to enhance parent-child relationships and reduce secrecy in online behaviors [47]. Commodari et al. [48] emphasized that balanced parental involvement, particularly parental monitoring, is crucial to reducing adolescents’ tendency to engage in video gaming as a means of escaping negative emotions, thereby mitigating the risk of excessive and problematic gaming. However, Görgülü and Özer [49] also caution that overly controlling parental mediation could unintentionally increase the risk of gaming disorder, suggesting that parental interventions should focus on autonomy-supportive rather than controlling strategies.

Adolescents who encounter suboptimal communication, inadequate emotional support, or elevated levels of conflict with their parents are predisposed to manifest problematic online behaviors, such as excessive social media utilization and compulsive browsing.

## 5. Conclusions

Problematic Internet Use is associated with ADHD symptoms. Individuals with ADHD who also exhibit problematic digital media use show more severe symptoms, greater difficulties with executive functions, a more dysfunctional family environment, increased pressure from negative life events, and less motivation for learn [31]. The complexity of the relationship between PIU, or IGD, and ADHD necessitate additional research to better understand the direction of the association and the underlying mechanisms. The mixed findings highlight the complex interplay between the inherent characteristics of online games, the consequences of excessive use, and the resulting psychopathologies [50].

A multifaceted strategy is needed to address these vulnerabilities, including interventions that support social and academic difficulties, increase self-regulation, and improve time management abilities. It is crucial to implement primary and secondary prevention interventions, especially in school settings, to promote healthy behaviors, increase awareness of digital media use, and support effective coping strategies to manage emotions and stress. Without such interventions, adolescents with ADHD may experience a worsening of their Internet addiction and IGD cycle.

Increased Internet and digital media engagement among adolescents and young adults necessitates urgent exploration of the psychological, educational, and social consequences associated with these behaviors [51,52,53].

## 6. Limitation and Future Research

This scoping review presents several limitations that may affect the breadth and comprehensiveness of its results. The search strategy employed solely a single digital database, i.e., Scopus, potentially resulting in the exclusion of relevant studies available in alternative databases, such as PubMed, WoS, or PsycINFO. Furthermore, some studies have been conducted on specific groups [16] such as adolescents who voluntarily sought evaluation [31]. It is acknowledged that these study samples may differ substantially from the general population, a factor that may potentially limit the generalizability and broader applicability of the findings. Moreover, the present scoping review did not differentiate specific types of impulsivity (e.g., cognitive versus motor impulsivity) that might differentially affect gaming behavior. It is recommended that future studies explore these distinct impulsivity dimensions to clarify their unique roles in the development and persistence of Problematic Internet Use and Internet Gaming Disorder in adolescents with ADHD.

It is recommended that future research address the methodological limitations identified by incorporating more inclusive search strategies and representative study samples. This would enhance the robustness and applicability of the evidence in this important area.

Given the complexity of issues related to ADHD and Internet use, it is essential to adopt multidisciplinary approaches involving various different professionals (psychologists, neuropsychiatrists, educators, social workers) to ensure comprehensive patient care.

Future research should focus on longitudinal studies to better understand the relationship between PIU, or IGD, and ADHD, as there are still many unanswered questions. While cross-sectional studies are useful, they cannot tell us whether IGD is caused by ADHD symptoms, whether ADHD raises the risk of developing IGD, or whether both disorders are influenced by common factors like executive function difficulties or psychological vulnerabilities.

For example, longitudinal studies can help clarify the direction of the association by tracking how ADHD symptoms evolve in people who develop IGD over time. Experimental and randomized controlled trial (RCT) methodologies could prove particularly valuable in investigating causal relationships between ADHD symptoms and PIU or IGD.

Future studies should investigate whether ADHD predisposes individuals to IGD, whether IGD exacerbates pre-existing ADHD symptoms, or whether both disorders are influenced by common etiological factors, such as deficits in executive functioning, difficulties in self-regulation, or psychological vulnerabilities [54,55].

Ultimately, ongoing research in this area is critical, given its substantial implications for public health, education, and clinical practice. Clearly defining unresolved questions and systematically investigating the complex interactions between ADHD and digital media use are essential for developing evidence-based guidelines for diagnosis, prevention, and intervention in this rapidly evolving field.

## Figures and Tables

**Figure 1 ijerph-22-00496-f001:**
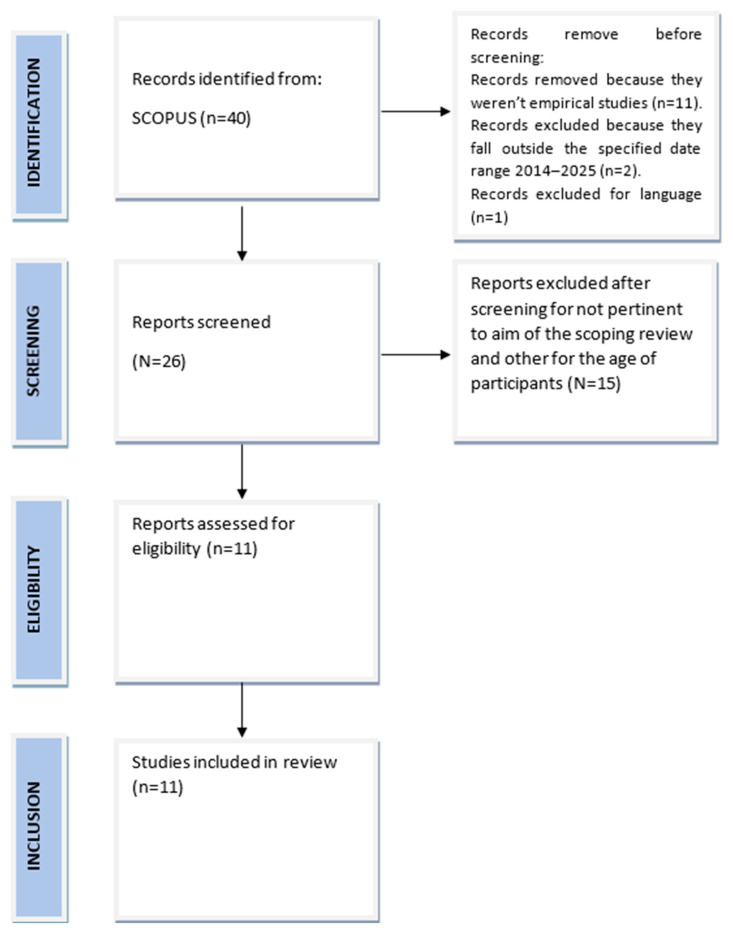
Flowchart of the studies analyzed in this review, according to Preferred Reporting Items for Systematic reviews and Meta-Analyses (PRISMA).

**Table 1 ijerph-22-00496-t001:** Summary of the reviewed studies.

Authors	Design	Sample	Aims	Outcomes	Findings
Özkan et al. [16]	Cross-sectional	97 adolescents (ages 13–18) with ADHD + 97 healthy controls	Investigated associations among ADHD symptom severity, metacognition, problematic social media use, and cyberbullying/cybervictimization in adolescents with ADHD.	Metacognitions, problematic social media use, cyberbullying, cybervictimization, ADHD symptom severity.	Adolescents with ADHD presented higher levels of dysfunctional metacognitions, problematic social media use, and increased cyberbullying/cybervictimization compared to controls. Specific metacognitions (e.g., positive meta-worry) correlated positively with ADHD symptoms, and problematic social media use was mainly predicted by inattention symptoms.
Seyrek et al. [36]	Cross-sectional	468 students (ages 12–17)	Assessed the prevalence of Internet Addiction (IA) and its associations with sociodemographic variables and psychopathological factors (depression, anxiety, ADHD).	Prevalence and severity of IA, correlations with depression, anxiety, ADHD symptoms, personal habits (e.g., smoking).	About 1.6% of participants met criteria for IA and 16.2% had possible IA. IA correlated significantly with depression, anxiety, ADHD symptoms, and smoking. There was no significant association with age, sex, BMI, school type, or socioeconomic status.
Gostoli et al. [17]	Multicenter, cross-sectional	440 first-year students (mean age ~14)	Explored how ADHD symptoms and psychosocial factors (allostatic overload, psychological wellbeing) impact unhealthy lifestyle behaviors (ULBs).	Substance use (alcohol, cigarettes, cannabis), sleep quality/quantity, problematic use of technological devices.	The relationship between ADHD symptoms and reduced sleep was stronger in cases of low allostatic overload and high levels of positive relationships. Furthermore, with higher environmental mastery, adolescents with ADHD spent more time on technological devices instead of sleeping.
Benedetto et al. [38]	Cross-sectional	676 Italian high school students (ages 15–19)	Examined the relationship between smartphone distraction, emotional/behavioral factors, and Internet Addiction risk.	The importance of addressing cognitive-emotional processes and individual vulnerability factors in preventing and managing Internet addiction among adolescents.	Higher IA levels were associated with greater smartphone distraction (emotional regulation, impulsive attention, online vigilance, multitasking) and emotional/behavioral problems (especially hyperactivity/inattention). Adolescents with problematic IA showed higher scores on all smartphone distraction dimensions and greater emotional/behavioral issues than non-problematic users.
Shuai et al. [31]	Cross-sectional (COVID-19 period)	192 children/adolescents with ADHD (ages 8–16)	Examined the impact of PDMU on ADHD symptoms, executive functions, family environment, and study motivation.	ADHD symptoms, emotional/behavioral problems, executive functions, life event stress, study motivation, digital media usage time.	ADHD children with PDMU showed higher levels of inattention, oppositional defiance, behavioral/emotional problems, anxiety/depression, more severe executive function deficits, and lower study motivation than those without PDMU. Enhanced supervision and increased physical exercise were recommended to manage these issues.
Chang et al. [33]	Cross-sectional, scale validation in a clinical setting	102 children/adolescents with ADHD (ages 7–18)	Validated the Taiwanese version of the Internet Gaming Disorder Scale–Short Form (IGD-SF-T-L) and proposed a diagnostic cutoff for IGD in ADHD.	Psychometric properties (reliability, construct validity) of IGD-SF-T-L, cutoff for IGD.	The IGD-SF-T-L demonstrated strong internal consistency and construct validity; a cutoff score ≥ 10 showed good diagnostic accuracy. ADHD youth with IGD had more severe symptoms, more comorbid conditions, and poorer interpersonal relationships than ADHD youth without IGD.
Jeong et al. [32]	Prospective 2-year cohort study	2319 students in grades 3, 4, and 7	Explored factors related to IGD severity, incidence, and persistence in a large youth sample.	IGD risk daily gaming time, game types, mental health symptoms, family/parent-child relationship, social support.	Independent risk factors for IGD incidence included ≥240 min/day of online gaming, multiplayer games, depressive symptoms, and ADHD symptoms. Factors associated with increased IGD severity included 60–239 min/day of gaming, single-player games, higher parental attachment, and social support. Persistence was predicted by ≥240 min/day of gaming and ADHD symptoms.
Lung et al. [39]	Analysis of national cohort data	17,694 adolescents (age 12) from the TBCS	Investigated the mediating role of dissociative (absorptive) trait in the association between childhood diagnoses (ADHD, LD, learning disabilities) and Problematic Internet Use.	Internet use duration at age 12, dissociative trait, ADHD, ASD, LD, ID diagnoses.	Children with ADHD and LD showed higher dissociative absorption traits, which in turn increased their risk of Problematic Internet Use. Spending more than 5 h online on weekends was associated with lower perceived happiness.
Hygen et al. [35]	Longitudinal study (ages 10, 12, 14)	702	Examined long-term relationships between IGD symptoms and psychiatric disorders (depression, anxiety, ADHD, ODD/CD).	IGD (Internet Gaming Disorder Interview), psychiatric symptoms (CAPA).	The co-occurrence of IGD and other psychiatric symptoms primarily stemmed from common underlying factors rather than a direct causal relationship. However, increased IGD symptoms predicted a small reduction in anxiety symptoms over time.
Gostoli et al. [37]	Multicenter cross-sectional	440 adolescents (mean age 14.21)	Investigated the prevalence of unhealthy lifestyle behaviors (ULBs), ADHD symptomatology, and related psychosocial factors.	ADHD symptoms (ASRS), unhealthy lifestyle behaviors, allostatic overload, psychological wellbeing.	Adolescents with subclinical ADHD symptoms exhibited unhealthy lifestyle behaviors and psychosocial impairment similar to those with clinical ADHD, including unhealthy eating, alcohol use, sleep problems, and problematic technology use.
Hsieh et al. [34]	Cross-sectional	231 parents (83% mothers) of adolescents (ages 11–18) with ADHD	Explored correlates of parental self-efficacy in managing adolescent Internet use among families with adolescents who have ADHD.	Parental self-efficacy, ADHD/ODD symptoms, parental depression, parenting behaviors, adolescent Internet addiction.	Greater ODD symptoms and higher Internet addiction in adolescents were associated with lower parental self-efficacy. By contrast, higher levels of parental care improved parents’ self-efficacy in managing adolescent Internet use.

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
