# Peer review of "Exploring the Association Between Problematic Internet Use, Internet Gaming Disorder in Adolescents with ADHD: A Scoping Review"

_ijerph, 2025, doi:10.3390/ijerph22040496_

Round 1

Reviewer 1 Report

Comments and Suggestions for Authors

Thank you for the opportunity to review the article entitled "Exploring the association between Problematic Internet Use, Internet Gaming Disorder in adolescents with ADHD: A scoping review". The study intends to systematically review the literature regarding the association between ADHD, IGD and PIU in adolescents, with the intent of being able to provide indications in the prevention and treatment fields. Although I believe that the topic is very interesting, I believe there are several points, especially methodological, that do not make the article eligible for publication in this journal. In particular, there is some confusion between IGD and PIU, both in relation to the literature and in the choice to put these constructs together. It is true that PIU also contains adolescents' gaming problems, but it also refers to many other problems (social networks, online pornography, etc.), which make the problem very heterogeneous and not useful in a more specific systematic review. Furthermore, the keywords used are “ADHD” and “adolescence,” combined with either “problematic use disorder” or “internet gaming disorder.” The use of these simple words leaves out many declinations of these terms that would have allowed to further broaden the analysis of the literature and, most likely, would have led to a greater analysis of the phenomenon (for example, "adolescent*", "Attention-Deficit/Hyperactivity Disorder", IGD, Gaming disorder* etc). Even more, the analysis within a single database (i.e., Scopus) makes the literature search very limited. A systematic review should analyze the titles emerging from the search on at least 3/4 databases. Furthermore, the results do not show a real analysis of similarities, differences, common points (e.g. tools used, differences emerging from a diagnostic point of view) but simply report the results of each study. I therefore believe that the work does not contribute more to the existing literature. Nevertheless, the focus is certainly of interest and if the authors set up the work with greater methodological rigor, it would certainly be of interest

Comments on the Quality of English Language

There are several errors in the use of English language, so a proper revision would be necessary

Author Response

Dear Reviewer,

We are immensely grateful for your invaluable time and effort in conducting a thorough review of our manuscript and offering constructive feedback. We acknowledge your concerns and would like to address them accordingly.

Firstly, we would like to clarify that our study was conducted as a scoping review, rather than a systematic review. We acknowledge that this distinction may not have been sufficiently emphasized in our manuscript and we will ensure that this aspect is clarified.

With regard to the database selection, we fully agree that limiting the search to Scopus constitutes a limitation of our study. We acknowledge that conducting a more extensive literature search across multiple databases (e.g., PubMed, Web of Science, PsycINFO) could have provided a more comprehensive overview. This is a significant aspect that will be emphasized in the limitations section, and it is intended to be addressed in future research by expanding the database selection.

Concerning the conceptual distinction between Problematic Internet Use (PIU) and Internet Gaming Disorder (IGD), we recognize the complexity of these constructs. Our objective was to examine the overlap between these constructs in adolescents with ADHD, as prior research indicates that ADHD is a risk factor for both. However, we acknowledge that further clarification is needed regarding how these terms were operationalized in our study, and we will improve this discussion in the revised manuscript.

We would also like to express our gratitude for your valuable suggestion concerning the selection of keywords. It is acknowledged that the use of more inclusive terms (e.g., "adolescent," "Attention-Deficit/Hyperactivity Disorder," "IGD," "Gaming Disorder") could have resulted in a more extensive dataset. In future studies, we will refine our search strategy to ensure broader coverage of relevant literature. Finally, we acknowledge your concerns regarding the presentation of results. However, we concur that accentuating the commonalities, differences, and methodological aspects (e.g., tools employed, diagnostic frameworks) could enhance the clarity and utility of our findings. We shall therefore revise the sections dealing with the results in order to place greater emphasis on these aspects.

We have carefully addressed your comments by making the necessary English language edits, which you can see marked in red. Furthermore, the additional changes related to the comments have been implemented, and are highlighted in yellow.

Yours sincerely,

Anna Passaro

Reviewer 2 Report

Comments and Suggestions for Authors

The present paper explores the relationship between Problematic Internet Use, Internet Gaming Disorder, and the presence of ADHD in adolescents. The topic of the study is interesting and is well developed by the authors.

However, there are some minor issues that need to be addressed.

1. The discussion needs to be further developed by specifying more clearly the results reported by the quoted authors and describing the test used and the samples involved.

2. In the conclusions, the importance of this research area should be emphasized, especially highlighting the possible future objectives based on the gaps in the current literature. Finally, the authors should better specify the open problems with the ongoing research in this field.

Comments on the Quality of English Language

The quality of English does not limit the understanding of the research.

However, there are some spelling and grammatical errors. I recommend a review of the English before the new submission.

Author Response

Dear Reviewer,

we appreciated your comments and revised accordingly. 

Please find our replies below. 

  1. In the Results section, we included a synoptic table summarizing the reviewed studies with the participants involved, the main objectives and outcomes.
  2. We added a final limitations and future research section addressing your concern.

The comments have been implemented and are highlighted light blue.

Best regards.

Anna Passaro

Reviewer 3 Report

Comments and Suggestions for Authors

Thank you for contributing to the growing body of research on ADHD and digital media use. The manuscript effectively summarizes findings on the intersection of ADHD, PIU, and IGD in adolescents. However, several key aspects require revision to improve clarity, methodological rigor, and engagement with the broader literature. Suggested major and minor revisions are provided below:

  1. The manuscript states that the search was conducted using the Scopus database. However, it is not specified whether other databases (e.g. PubMed, PsycINFO) were considered. Given the interdisciplinary nature of the topic, it would be helpful to justify why only Scopus was chosen.
  2. Inclusion and exclusion criteria should be specified. In particular, were studies excluded because of sample size, flaws in study design, or low methodological quality? Although not mandatory for scoping reviews, an assessment of the risk of bias would certainly add rigor to the study.
  3. The discussion largely reiterates the findings without providing a critical assessment of the limitations of the existing research. For example, while impulsivity and emotional regulation are highlighted as key risk factors, the manuscript does not adequately address how different types of impulsivity (e.g., cognitive versus motor impulsivity) might differentially affect gaming behavior.
  4. Recommendations for future research should be more specific. Rather than suggesting longitudinal studies in general, consideration should be given to discussing specific methodologies (e.g., experimental designs or interventions targeting ADHD symptoms to reduce PIU).
  5. The paper discusses the role of parenting styles and social influences in adolescent gaming behavior. I recommend citing recent studies on how parental mediation strategies can mitigate excessive gaming behaviors in adolescents to strengthen the argument of the discussion (e.g., 10.1111/jora.13034; 10.1007/s12144-023-04557-6; 10.3390/bs13080679).
  6. A summary table comparing the methodologies of the studies reviewed (e.g., sample size, country, research design) would provide a clearer overview for readers.
Comments on the Quality of English Language

Some sentences are overly complex and should be revised to make them clearer.

Author Response

Dear Reviewer,

we are grateful for your comments and suggestions. Here by we reply point by point.

  1. We searched in Scopus as the largest database of Literature. We aknowledged in the limitations section that no further databases were used. We will consider your suggestion for a future systematic review.
  2. Both including and excluding criteria were detailed in the original version of the manuscript. Additionally, the PRISMA flow addressed your concern on the excluded studies.
  3. We further argued the discussion and included a final limitations and future research perspectives addressing your concern. 
  4. We addressed your suggestion on the final limitations and future research section. 
  5. We included the references you suggested. 
  6. We included a synoptic table with the reviewed studies. 

We green highlighted the modified sections throughout. We have carefully addressed your comments by making the necessary English language edits, which you can see marked in red. 

Yours sincerely,

Anna Passaro

Round 2

Reviewer 1 Report

Comments and Suggestions for Authors

I thank the authors for their response and the changes made. The limitations of the study have been highlighted in the appropriate section, showing this study as a review of the underlying problem, which should however be investigated in more rigorous studies

Reviewer 3 Report

Comments and Suggestions for Authors

No further revisions are requested.